# Extended Sentinel Monitoring of *Helicoverpa zea* Resistance to Cry and Vip3Aa Toxins in Bt Sweet Corn: Assessing Changes in Phenotypic and Allele Frequencies of Resistance

**DOI:** 10.3390/insects14070577

**Published:** 2023-06-25

**Authors:** Galen P. Dively, Tom P. Kuhar, Sally V. Taylor, Helene Doughty, Kristian Holmstrom, Daniel O. Gilrein, Brian A. Nault, Joseph Ingerson-Mahar, Anders Huseth, Dominic Reisig, Shelby Fleischer, David Owens, Kelley Tilmon, Francis Reay-Jones, Pat Porter, Jocelyn Smith, Julien Saguez, Jason Wells, Caitlin Congdon, Holly Byker, Bryan Jensen, Chris DiFonzo, William D. Hutchison, Eric Burkness, Robert Wright, Michael Crossley, Heather Darby, Tom Bilbo, Nicholas Seiter, Christian Krupke, Craig Abel, Brad S. Coates, Bradley McManus, Billy Fuller, Jeffrey Bradshaw, Julie A. Peterson, David Buntin, Silvana Paula-Moraes, Katelyn Kesheimer, Whitney Crow, Jeffrey Gore, Fangneng Huang, Dalton C. Ludwick, Amy Raudenbush, Sebastian Jimenez, Yves Carrière, Timothy Elkner, Kelly Hamby

**Affiliations:** 1Department of Entomology, University of Maryland, College Park, MD 20742, USA; 2Department of Entomology, Virginia Tech, Blacksburg, VA 24060, USA; 3Department of Entomology, Virginia Tech, Suffolk, VA 23434, USA; 4Virginia Tech ESAREC/Entomology, Painter, VA 23420, USA; 5Pest Management Office, Rutgers University, New Brunswick, NJ 08901, USA; 6LIHREC, Cornell University, Riverhead, NY 11901, USA; dog1@cornell.edu; 7Department of Entomology, Cornell AgriTech, Geneva, NY 14456, USA; 8Rutgers Agricultural Research and Extension Center, Rutgers University, Bridgeton, NJ 08302, USA; 9Department of Entomology and Plant Pathology, NC State University, Raleigh, NC 27601, USA; 10Department of Entomology and Plant Pathology, NC State University, Plymouth, NC 27962, USA; 11Department of Entomology, Penn State University, University Park, PA 16802, USA; 12Cooperative Extension, Carvel REC, University of Delaware, Georgetown, DE 19947, USA; 13Ohio Agricultural Research and Development Center, Wooster, OH 44691, USA; 14Department of Plant and Environmental Sciences, Clemson University, Florence, SC 29501, USA; 15Department of Entomology, AgriLife Research and Extension Center, Texas A&M University, Lubbock, TX 79401, USA; 16Department of Plant Agriculture, University of Guelph, Ridgetown Campus, ON N1G 2W1, Canada; 17CEROM, 740 Chemin Trudeau, Saint-Mathieu-de-Beloeil, QC J3G 0E2, Canada; 18New Brunswick Department of Agriculture, Sussex, NB E4E 5L8, Canada; 19Perennia Food and Agriculture, Kentville, NS B4N 1J5, Canada; 20Department of Plant Agriculture, University of Guelph, Winchester, ON N1G 2W1, Canada; 21Arlington Agricultural Research Station, University of Wisconsin, WI 53706, USA; 22Department of Entomology, Michigan State University, East Lansing, MI 48824, USA; 23Department of Entomology, University of Minnesota, St. Paul, MN 55455, USA; 24Department of Entomology, University of Nebraska-Lincoln, NE 68588, USA; 25Department of Entomology and Wildlife Ecology, University of Delaware, Newark, DE 19711, USA; 26Department of Plant and Soil Sciences, University of Vermont, Burlington, VT 05405, USA; 27Department of Plant and Environmental Sciences, Clemson University, Charleston, SC 29414, USA; 28Illinois Extension, University of Illinois, Urbana, IL 61820, USA; 29Department of Entomology, Purdue University, West Lafayette, IN 47906, USA; 30USDA-ARS Corn Insects and Crop Genetics Research, Iowa State University, Ames, IA 50011, USA; 31South Dakota State, Brookings, SD 57006, USA; 32Panhandle Research and Extension Center, Scottsbluff, NE 69361, USA; 33West Central Research and Extension Center, University of Nebraska, North Platte, NE 69101, USA; 34Griffin Campus, University of Georgia, Griffin, GA 30223, USA; 35UF/IFAS West Florida Research and Education Center, Jay, FL 32565, USA; 36Department of Entomology & Plant Pathology, Auburn University, Auburn, AL 36849, USA; 37Department of Biochemistry, Molecular Biology, Entomology and Plant Pathology, Delta Research and Extension Center, Mississippi State University, Stoneville, MS 39762, USA; 38Department of Entomology, Louisiana State University, Baton Rouge, LA 70803, USA; 39Department of Entomology, Texas A&M AgriLife Extension Service, Corpus Christi, TX 78404, USA; 40PEI Department of Agriculture and Land, Charlotte, PE C1A 7N8, Canada; 41Department of Entomology, University of Arizona, Tucson, AZ 85721, USA; 42Southeast Research and Extension Center, Landisville, PA 17538, USA

**Keywords:** corn earworm, bollworm, resistance monitoring, *Bacillus thuringiensis* toxins, phenotypic and allele resistance frequency

## Abstract

**Simple Summary:**

Corn and cotton that produce insecticidal toxins derived from *Bacillus thuringiensis* (Bt) are widely adopted in the United States to control corn earworm/cotton bollworm, *Helicoverpa zea* (Boddie), which has resulted in major benefits to growers and the general public. However, resistance evolution in *H. zea* populations has become a major threat to the sustainability of these crops. Bt sweet corn producing the same toxins as Bt field corn is more attractive to *H. zea* than field corn and, thus, can function as a sentinel plant to detect early stages of resistance. As part of an existing sentinel monitoring network, this study evaluated changes in *H. zea* resistance during 2020–2022 by estimating the phenotypic and resistance allele frequencies for toxins in sentinel Bt corn.

**Abstract:**

Transgenic corn and cotton that produce Cry and Vip3Aa toxins derived from *Bacillus thuringiensis* (Bt) are widely planted in the United States to control lepidopteran pests. The sustainability of these Bt crops is threatened because the corn earworm/bollworm, *Helicoverpa zea* (Boddie), is evolving a resistance to these toxins. Using Bt sweet corn as a sentinel plant to monitor the evolution of resistance, collaborators established 146 trials in twenty-five states and five Canadian provinces during 2020–2022. The study evaluated overall changes in the phenotypic frequency of resistance (the ratio of larval densities in Bt ears relative to densities in non-Bt ears) in *H. zea* populations and the range of resistance allele frequencies for Cry1Ab and Vip3Aa. The results revealed a widespread resistance to Cry1Ab, Cry2Ab2, and Cry1A.105 Cry toxins, with higher numbers of larvae surviving in Bt ears than in non-Bt ears at many trial locations. Depending on assumptions about the inheritance of resistance, allele frequencies for Cry1Ab ranged from 0.465 (dominant resistance) to 0.995 (recessive resistance). Although Vip3Aa provided high control efficacy against *H. zea*, the results show a notable increase in ear damage and a number of surviving older larvae, particularly at southern locations. Assuming recessive resistance, the estimated resistance allele frequencies for Vip3Aa ranged from 0.115 in the Gulf states to 0.032 at more northern locations. These findings indicate that better resistance management practices are urgently needed to sustain efficacy the of corn and cotton that produce Vip3Aa.

## 1. Introduction

Transgenic corn and cotton that produce insecticidal toxins derived from *Bacillus thuringiensis* Berliner (Bt) are widely used to control multiple insect pests in the United States and other countries [1]. Bt crops have reduced yield loss, insecticide use, and non-target effects, resulting in environmental, human health, and economic benefits to growers and the general public [2,3,4,5,6,7,8,9,10,11].

Resistance evolution in target insect populations is a major concern due to the selection pressure exerted by the high constitutive expression of Bt toxins throughout the crop cycle. Accordingly, the U.S. Environmental Protection Agency (EPA) values plant-incorporated Bt toxins as a public good and mandates insect resistance management (IRM) plans as part of the commercial registration of Bt crops to maintain their sustainability [12,13]. IRM best practices include a high dose expression of the Bt toxins in the crop, to prevent the survival of offspring born from the mating between susceptible and resistant individuals, together with structured, seed blended, or natural refuges of non-Bt plants that produce susceptible individuals to mate with resistant ones, thus reducing the resistance allele frequency [14,15,16]. Additionally, most Bt corn and Bt cotton plants now produce two or more pyramided toxins that redundantly target the same pest to slow the evolution of resistance [17,18,19].

To detect resistance and implement mitigation measures before control failures occur, industry registrants of Bt crops are required to monitor the evolution of resistance in target pest populations [13,20,21]. For lepidopteran pests, the monitoring approach used by registrants consists of discriminating dose bioassays of larvae collected from non-Bt host plants in major production areas and investigations of unexpected pest damage in Bt crop fields [13]. Since Bt corn was first commercialized in 1996, monitoring efforts by registrants have not detected any decreases in susceptibility to Bt toxins in European corn borer populations (*Ostrinia nubilalis* (Hübner) Lepidoptera: Crambidae). However, the first case of resistance to the Cry1F toxin in Bt corn was confirmed from Nova Scotia populations in 2018 [22], and a significant resistance to other Cry toxins was recently reported from field-collected populations in other Canadian provinces [23]. The fall armyworm (*Spodoptera frugiperda* (J.E.Smith) Lepidoptera: Noctuidae) has been targeted by the Cry1F toxin since 2003, without any evidence of field-evolved resistance until 2010 when widespread control failures in Bt corn were reported in Puerto Rico and when, later, F2 screen studies showed high levels of resistance in four southeastern U.S. states [24]. Early resistance monitoring studies and reports by registrants did not show strong evidence of significant changes in the baseline level of susceptibility to Bt toxins in corn earworm/cotton bollworm (*Helicoverpa zea* (Boddie) Lepidoptera: Noctuidae) populations [25,26,27]. However, more recent studies demonstrate widespread field-evolved resistance in *H. zea* to Cry1Ab, Cry1Ac, Cry2Ab2, and Cry1A.105 in Bt corn and Bt cotton [28,29,30,31,32,33,34,35,36,37,38,39]. *Helicoverpa zea* is not highly susceptible to Cry toxins, implying that key conditions underlying the success of the refuge strategy (i.e., high dose, recessive inheritance, and complete redundancy killing) for Bt crops producing these toxins may not be met [40,41,42]. Other factors contributing to resistance include a reduced refuge size or the absence of refuges, an initial frequency of resistance alleles, dispersal and mating behavior, cross resistance between Cry toxins, increased selection pressure from pyramided toxin expression, and selection in multiple generations across multiple Bt crops [18,30,43,44,45,46,47]. In contrast, Vip3Aa in Bt corn and cotton still provides excellent control of *H. zea* under field conditions [29,34,48,49], although there is growing evidence that there is a high risk of resistance evolution to Vip3Aa in *H. zea*, particularly in the southeastern U.S. [34,50,51,52,53].

More effective monitoring approaches are clearly needed to identify resistance early enough to enable proactive mitigation measures [44,46,54]. Fritz [55] reviewed the successes and limitations of genomic methods to detect and understand mechanisms of insect resistance, but further advancements are likely needed before they can provide an adequate warning of early stages of resistance to Bt toxins. Several EPA scientific advisory panels [56,57,58,59] have addressed the limitations and challenges of the diet bioassay approach used by registrants, as well as the utility of sentinel plot monitoring. Reisig et al. [46] proposed five best management practices to delay lepidopteran resistance to Bt crops, including the sentinel plot monitoring of pest survival and damage to detect practical resistance [21]. Venette et al. [60] proposed that sentinel Bt sweet corn planted side-by-side with its non-Bt isoline can function as an in-field diagnostic screen to monitor changes in control efficacy and the phenotypic frequency of resistance (PFR), defined as the ratio of larval densities in Bt ears relative to densities in non-Bt ears. Using this approach, a significant reduction in control efficacy coupled with an increased PFR can be viewed as a genetically based change in susceptibility and a confirmation of field-evolved resistance [21,61]. Dively et al. [29] used paired Bt and non-Bt sweet corn plots in Maryland to track changes in *H. zea* susceptibility and reduced control efficacy as evidence of field-evolved *H. zea* resistance to Cry1Ab and pyramided CryA.105/Cry2Ab2 toxins. As a continuation of this approach, an expanded monitoring network of sweet corn sentinel trials was implemented during 2017–2019, which reported significant reductions in the control efficacy of Cry toxins and a possible decrease in *H. zea* susceptibility to Vip3Aa [34]. This study also outlined the strengths and limitations of the field-based sentinel approach compared to a laboratory-based diet bioassay for monitoring resistance and recommended improvements in the design of sentinel plot monitoring.

Here, we evaluated further changes in the phenotypic resistance in *H. zea* to Cry1Ab and the pyramided toxins of Cry1A.105/Cry2Ab2 and Cry1Ab/Vip3Aa expressed in sentinel sweet corn during 2020–2022. Compared to our previous work [34], we expanded the network to include more trial locations and included multiple plantings per location, particularly in the North Central and Southern states; increased the sampling effort to detect early resistance evolution to the Vip3Aa toxin; and improved the timing of ear sampling to more accurately estimate the phenotypic frequency of resistance. Using the PFR ratios, we estimated the range of allele frequencies for resistance to Cry1Ab and Vip3Aa. Since we assumed the worst-case scenario that any live larvae associated with kernel damage in a Bt ear may indicate some level of resistance to the expressed toxins, our resistance frequency values are likely overestimated but still provide evidence of relative changes in resistance to the single or pyramided Bt toxins compared to published studies and previous sentinel monitoring results. Our sentinel trials also continued to simultaneously monitor for susceptibility changes and regional differences in other pests of corn, including *O. nubilalis*, *S. frugiperda*, and western bean cutworm (*Striacosta albicosta* (Smith) (Lepidoptera: Noctuidae)).

## 2. Materials and Methods

### 2.1. Sentinel Trial Locations

On a volunteer basis without outside funding, collaborators established 41 trials in 2020, 52 trials in 2021, and 53 trials in 2022, located in twenty-six states and five Canadian provinces (Figure 1). Solid circles indicate locations where *H. zea* successfully overwinters, and open circles indicate where *H. zea* populations are mainly established every year by migrant moths. In northern locations, trials were planted to synchronize silking with peak *H. zea* infestations resulting from migrant moths from southern populations that most likely were pre-selected in Bt field corn. Most collaborators established one trial each year; however, in 12 states, ON, and NS, they planted multiple trials at different times and/or locations. Particularly, multiple plantings in the Gulf states were timed to monitor the susceptibility of overwintered and summer populations that could be exposed to different selection regimes, depending on the source host plant.

### 2.2. Hybrids, Planting Arrangement, and Plot Size

Sentinel plots consisted of five sweet corn hybrids: Attribute ‘BC0805′ expressing Cry1Ab, Attribute II ‘Remedy’ expressing Cry1Ab and Vip3Aa, and their near non-Bt isohybrid ‘Providence’ (Syngenta Seeds); and Performance Series ‘Obsession II’ expressing Cry1A.105/Cry2Ab2 and its non-Bt isohybrid ‘Obsession I’ (Bayer–Seminis Seeds). Most trials included all five hybrids; however, since *H. zea* has already evolved high levels of resistance to the Cry toxins, some trials included only plantings of Remedy and Providence sweet corn to monitor changes in susceptibility to the Vip3Aa toxin.

Trials consisted of 4–8 rows of each hybrid, at least 15–30 m long, planted side-by-side. To minimize outcrossing between the sugar-enhanced and supersweet Bt hybrids, the plot layout consisted of the two non-Bt hybrids planted together in the center to act as a buffer, Remedy and BC0805 on one side, and Obsession II on the other side, preferably downwind from the prevailing winds. The Remedy plot was planted first, followed by the other Bt hybrids, and, lastly, the non-Bt isolines to reduce the risk of non-Bt seeds unintentionally being planted in Bt plots. Any remaining seed from the planter was carefully removed before starting and after planting each hybrid. Plots were planted at a seeding rate of 54,455 plants per ha and managed according to commercial production practices, including pre-plant and side-dressed fertilizer applications, residual herbicides, and irrigation to ensure normal plant growth. No foliar insecticides were applied unless pre-silk applications were needed to control a high *S. frugiperda* infestation.

### 2.3. Ear Sampling

Ear sampling in the Bt and non-Bt plots was timed to record the highest number of surviving *H. zea* larvae causing kernel injury. Ideally, ear sampling of non-Bt and Bt plots was conducted at different times. Non-Bt plots were sampled first around 18–21 days after the onset of silking, and then, Bt plots were sampled 5–6 days later to account for the delayed development of intoxicated larvae. However, most collaborators only sampled the plots once due to labor and time constraints, so ear sampling was delayed until more than 50% of the larvae were older instars in Cry1Ab and Cry1A.105/Cry2Ab2 plots. Although this sampling schedule allowed more time for a higher number of surviving larvae to reach older instars in the Bt ears, many mature larvae had already exited ears of non-Bt plants at this time. Consequently, data adjustments were made to account for missing larvae (explained below).

Under high *H. zea* infestations, a minimum sample of 50 primary ears from the center rows was evaluated to assess the level of kernel damage and larval stages in the non-Bt plots. For Bt hybrids, 100–200 ears per plot were sampled, and, generally, larger samples were taken from the Cry1Ab/Vip3Aa hybrid because it was mostly free of kernel injury and larvae. Altogether, the number of ears sampled per hybrid over the three years totaled 8234 (non-Bt Providence), 9104 (Cry1Ab BC0805), 6885 (non-Bt Obsession I), 8119 (Cry1A.105/Cry2Ab2 Obsession II), and 15,563 (Cry1Ab/Vip3Aa Remedy). Each ear was carefully opened at the tip to expose larvae and kernel injury, and husk leaves were occasionally removed all the way to expose the base of the ear if there were signs of entry or exit on the sides and shank caused by *O. nubilalis*, *S. frugiperda*, or *S. albicosta*.

### 2.4. Recorded Data

The following information was recorded for individual ears: number of each larval instar (alive or dead) by species, location of feeding injury (either on silk tissue, ear tip, or upper or lower half of the ear), kernel area consumed, and presence of exit holes. On each ear, the entire area of kernel injury, which often included overlapping feeding by several larvae, was visually evaluated to estimate the total cm^2^ of kernel area consumed. A convenient reference was the cross-section area of a standard pencil eraser (0.5 cm^2^). Several collaborators also overlaid the damaged area with a transparent sheet outlined in a 0.25 cm^2^ grid as a visual guide. For ears with very minor feeding injury (<0.5 cm^2^) on a few kernels on the ear tip, 0.25 cm^2^ was recorded to indicate that the ear was successfully invaded by larvae and that some kernel feeding occurred. This was characteristic of the majority of *H. zea* damage reported for Vip3Aa ears, which were often associated with missing or dead early instars. Damage ≥ 0.5 cm^2^ was recorded to the nearest 0.5 cm. The same procedure was followed when recording *S. frugiperda* and *S. albicosta* kernel injury if it could be separated from *H. zea* injury. Ear injury caused by *O. nubilalis* was more difficult to delineate, particularly in non-Bt ears with extensive *H. zea* injury, and these larvae often tunneled into the cob. If *O. nubilalis* was present but the extent of its kernel consumption could not be determined, then 0.5 cm^2^ was recorded to acknowledge its presence.

### 2.5. Data Adjustments and Analysis

To calculate PFR and related metrics for each Bt hybrid, it was important to account for the highest number of *H. zea* larvae that survived to cause kernel injury, including those that had already exited ears. Therefore, specific information on each damaged Bt and non-Bt ear was carefully examined to determine if any recorded instars were old enough to account for the kernel area consumed. One *H. zea* larvae consumes 6–8 cm^2^ of kernel area during its development from 2nd instar to pupal stage (GPD, unpublished data). For ears with a kernel consumption pattern characteristic of *H. zea* feeding but without accountable larvae, the following data adjustments were made. One exited 6th instar was added to the data if kernel area consumed was ≥6 but <12 cm^2^, located on the tip and upper half of the ear, and associated with an exit hole or frass deposits characteristic of a 6th instar. In cases of extremely damaged ears without accountable larvae, two exited 6th instars were added if the kernel area consumed ≥12 cm^2^ and there was evidence of exit holes and separate characteristic feeding patterns on each side of the upper ear. Otherwise, no adjustment was made if kernel damage was <6 cm^2^ per ear without accountable larvae present or exit holes. The adjustment criteria were applied to both Bt and non-Bt damaged ears.

After data adjustments, the percentage of ears damaged by each species was averaged over both non-Bt hybrids to show the range of infestation levels among trial locations and monitoring years. We then compiled individual trial data for *H. zea* and computed the following metrics by year and hybrid: mean kernel area consumed per damaged ear, mean number of live larvae per ear, and proportion of late instars (fourth, fifth, and sixth). Overall means and standard deviations for each metric were calculated using the pooled data for all trials. Using a paired *t*-test (assuming uneven variances and a one-tailed hypothesis), we tested for significant differences in each metric between Bt and non-Bt isogenic pairs of hybrids, both within and between monitoring years.

The PFR in the *H. zea* population associated with each Bt hybrid was estimated for each individual trial as the ratio of mean density of surviving *H. zea* larvae (including those that exited, as described previously) per Bt ear relative to the mean surviving larvae per non-Bt ear. The accuracy of PFR as an in-field screen to detect resistance depended on the *H. zea* infestation level and whether ear sampling was properly timed to measure the highest number of larvae surviving in both Bt and non-Bt ears [60]. Therefore, we used a selected subset of trial data each year to estimate the overall mean PFR and 95% confidence limits for each Bt hybrid. The data subsets included only trials with *H. zea* infestations that caused damage on >50% of the Cry and non-Bt ears and consisted of >50% late instars.

Venette et al. [60] developed a method to estimate the PFR, the corresponding resistance allele frequency (RAF), and confidence intervals associated with PFR and RAF for sentinel plots planted in single-toxin Bt crops. This approach is not appropriate for estimating RAF for Bt crops producing two toxins that differ, importantly, in their amino acid sequences (e.g., Cry1Ab/Vip3Aa and Cry1A.105/Cry2Ab2 corn considered here) as evolution of resistance to such two-toxin crops is expected to involve mutations at two resistance loci [18,62]. However, the estimated RAF for Cry1Ab corn was 1 or close to 1 across the regions investigated here (see Results and Discussion), implying that resistance to Cry1Ab had no or little effects on variation in PFR for Cry1Ab/Vip3Aa corn. Accordingly, we assumed that resistance to Vip3Aa was monogenic, autosomal, and recessive [63] and used the mean PFR for Cry1Ab/Vip3Aa corn to estimate RAF for Vip3Aa. Because the frequency of alleles conferring resistance to Cry1A.105 and Cry2Ab2 was likely less than 1 (see Results and Discussion), we could not estimate RAF for either of these toxins. Using the relationships between PFR and RAF outlined by Venette et al. [60], the lower and upper limits of the confidence interval for RAF was computed as PFR/2 (if resistance was fully dominant) and √PFR (if resistance was fully recessive).

## 3. Results and Discussion

### 3.1. Occurrence and Infestation Levels of the Major Lepidopteran Larvae

In addition to resistance monitoring, our expanded network of more trial locations allowed for an assessment of the major lepidopteran larvae of corn over a larger geographical area. *Helicoverpa zea* ear infestations reached very high levels at most trial locations, causing an overall average of 7.17 cm^2^ of kernel consumption in 75.2%, 66.0%, and 75.7% of all non-Bt ears sampled in 2020, 2021, and 2022, respectively (Appendix A (online only)). Overall, 109 of the 146 trials reported larval infestations and kernel consumption in more than 50% of the non-Bt ears. The highest levels of infestations occurred at the southeastern and mid-Atlantic locations where successful *H. zea* overwintering occurs, whereas the lowest levels of infestations were mainly recorded in the North Central and Northeast states and Canadian provinces, where populations are mainly sourced by migrant moths. The overall levels of ear infestations and larval densities in Cry-expressing ears were slightly lower relative to the non-Bt isolines but were only significantly lower in the Cry1A.105/Cry2Ab2 hybrid relative to its non-Bt isoline (Table 1). As previously reported [34], Cry toxins in Bt sweet corn have lost more than 80% of their control efficacy against *H. zea* compared to when first commercialized. Over all trials, it is noteworthy that *H. zea* was the only ear-invading lepidopteran pest found alive in sweet corn ears expressing Cry1Ab/Vip3Aa.

As a primary target pest of Bt corn, the management of *O. nubilalis* has been highly successful [10,59] due to the high dose expression of the Cry toxins, the movement interchange between corn and surrounding natural refugia, and the fitness costs that likely contributed to delaying the evolution of resistance [64,65,66,67]. *Ostrinia nubilalis* feeding injury in non-Bt ears was recorded in only 30 of the 146 trials and associated with either missing or very few live larvae. The overall percentage of ears injured by *O. nubilalis* averaged 1.0%, 1.2%, and 0.7% in 2020, 2021, and 2022, respectively. Trial locations with consistent year-to-year *O. nubilalis* infestations (up to 27.5% damaged ears) included Painter, VA; Rock Springs, PA; Lancaster, PA; Pittstown, NJ; Riverhead, NY; and Geneva, NY, where the surrounding landscape likely contained relatively less Bt field corn acreage. The absence of *O. nubilalis* infestations was also reported at most sentinel trial locations during 2017–2019 [34] and concurs with reports of the areawide suppression of populations due to the high adoption of Bt field corn [4,10]. More importantly, we found no evidence of *O. nubilalis* larval survival or feeding injury in a total of 32,786 ears examined from the Bt sweet corn plots.

Similarly, we recorded infestations of *S. frugiperda* in 2.1% of all non-Bt ears sampled and in only 41 of the 146 trials. Consistently with previous monitoring network results [34], ear infestations varied widely across trial locations and monitoring year, likely depending on the seasonal recruitment of *S. frugiperda* populations in the south and the frequency and direction of storm fronts that enabled migrant moths to reach northern locations. The percentage of damaged non-Bt ears was highest in Painter, VA (45.0%), Florence, SC (53.0%), and Lafayette, IN (11.0%), in 2020; Lubbock, TX (16.0%) and Corpus Christi, TX (20.4%), in 2021; and Lubbock, TX (20.0%), in 2022. Although our data on this pest are limited, Cry1A.105/Cry2Ab2 appeared to be more effective against *S. frugiperda* than Cry1Ab. Over the three years, 6 trials reported up to 6% of Cry1A.105 + Cry2Ab2 ears injured by *S. frugiperda*, compared to 21 trials with up to 32% of damaged Cry1Ab ears.

Ear infestations of *S. albicosta* larvae were uncommon as we only recorded them in 1.4% of all non-Bt ears sampled and in 14 of the 146 trials, which were located in NE, MI, and the Canadian Provinces. Generally, the control efficacy of the Bt hybrids against *S. albicosta* was lower and showed no consistent difference between the single and dual Cry-expressing hybrids compared to the control efficacy against other lepidopteran pests.

### 3.2. Ear Damage, Larval Development, and Kernel Consumption by H. zea

Overall, there were no significant or consistent trends in the percentage of damaged ears, number of larvae per ear, kernel area consumed, and proportion of late instars over the three years for the Cry1Ab (BC0805), Cry1A.105/Cry2Ab2 (Obsession II), and their non-Bt isohybrids (Table 1). However, there were significant differences between each pair of isogenic Bt and non-Bt hybrids. Only kernel consumption in 2020 and the proportion of late instars in 2020 and 2022 were significantly lower in the Cry1Ab hybrid than its non-Bt Providence. During all years, ear damage, larval density per ear, larval age, and kernel consumption were significantly lower for the Cry1A.105/Cry2Ab2 hybrid than for its non-Bt Obsession I (paired *t*-test, *p* < 0.05). The overall suppression of *H. zea* ear infestations by Cry1A.105/Cry2Ab2 was consistently higher than that of Cry1Ab. Pooled over years, the percentage of damaged ears, larval density, and kernel consumption were reduced by 15%, 19%, and 34% in the Cry1A.105/Cry2Ab2 ears and by 5%, −2%, and 17% in the Cry1Ab ears, respectively. Most notably, the mean number of larvae surviving in Cry1Ab ears was slightly higher than the mean number of larvae surviving in the non-Bt isogenic ears.

The trial data for Cry1Ab/Vip3Aa (Remedy) plots were highly skewed, with either no live larvae of any species or no kernel damage recorded on 99% of the sampled ears. Over all trials, *H. zea* was the only ear-feeding lepidopteran pest found alive on Cry1Ab/Vip3Aa ears. Moreover, all measurements of ear damage, larval development, and kernel consumption were significantly lower on Cry1Ab/Vip3Aa ears than on non-Bt isogenic ears (Table 1). Of 15,563 Cry1Ab/Vip3Aa ears sampled during 2020–2022, 156 ears (0.77%) had minor damage (<0.5 cm^2^, primarily on the tip), and only 25 of these ears (0.12%) were infested with a total of eighty-two live larvae (78% early instars). Most cases of ear damage and the presence of older larvae in Vip3Aa ears were from southern locations (TX, LA, MS, AL, and NC).

### 3.3. Phenotypic Resistance and Estimated Range of Resistance Allele Frequency

The individual estimates of PFR were based on 92 of the 127 trials with Cry1Ab data, 77 of the 132 trials with Cry1A.105/Cry2Ab2 data, and 116 of the 146 trials with Cry1Ab/Vip3Aa data (Appendix A (online only)). Trials with no or very low *H. zea* infestations or that were sampled too early to record the highest number of surviving larvae were excluded, and these were mostly from northern locations. The mean PFR ratios of *H. zea* larvae surviving in ears of isogenic Bt and non-Bt hybrids revealed no consistent trends over the three years (Table 2). PFRs for Cry1A.105/Cry2Ab2 were consistently lower than estimates for Cry1Ab. However, year-to-year differences between the Cry hybrids were not statistically significant, except for the PFR of 1.02 in 2022 for Cry1Ab, which was significantly higher than the 2022 estimate of 0.82 for Cry1A.105/Cry2Ab2, based on non-overlapping 95% confidence limits. In contrast, PFRs for Cry1Ab/Vip3Aa were significantly lower (ranging from 0.002 to 0.009) and not statistically different among the three years. Altogether, only 24 of the 119 trials with Cry1Ab/Vip3Aa had PFRs > 0, and most of these trials were located in the Gulf states.

The single Cry1Ab toxin in sweet corn and field corn was commercially introduced in 1996 and provided the consistent suppression of *H. zea* infestations until around the mid-2000s [68,69,70,71] when levels of phenotypic resistance began to increase [28]. Pooled over multiple plantings of paired plots of Cry1Ab and non-Bt sweet corn conducted in Maryland [29], the estimated PFR for Cry1Ab averaged 0.28 during 1996–2003, 0.61 during 2004–2010, and 0.67 during 2011–2016. Subsequently, sentinel sweet corn monitoring in 16 U.S. states and 4 Canadian provinces [34] reported increases in the PFR averaging 0.99 in 2017, 0.85 in 2018, and 0.76 in 2019 for Cry1Ab. Our results from 2020–2022 indicate that the Cry1Ab PFR may have plateaued over this study, ranging from 0.97 to 1.02 and averaging an overall 0.99 (Table 2). We also found no statistically significant changes over years in the percentage of damaged ears, kernel consumption, larval density, and instar development in the Cry1Ab hybrid. Moreover, there were no differences compared to the non-Bt isohybrid, except for slightly lower kernel consumption in Cry1Ab ears in 2020 only. The PFR was consistently higher at southern trial locations (mean = 1.04) where *H. zea* successfully overwinters but was not statistically lower at northern trials (mean = 0.97) where populations are sourced by migrant moths. Other studies and EPA reports have documented significant reductions in control efficacy in Bt field and sweet corn expressing Cry1Ab, along with relatively high resistant ratios and allele frequencies in field-collected populations [31,37,51,58,67,72,73,74].

The most disconcerting finding about Cry1Ab is that 49% of the individual trial values of PFR exceeded a ratio of 1, meaning that the number of *H. zea* surviving per Bt ear was higher than the number of larvae surviving per non-Bt ear. This was previously reported [34] and presumed to be the result of cannibalistic behavioral changes in larvae receiving sublethal doses of Cry1Ab. Although early instars of *H. zea* initially feed freely together in an ear, they become aggressively cannibalistic once they reach the 4th instar, and, thus, only one mature larva is often found remaining in a non-Bt ear. Sublethal intoxication by Cry1Ab is known to inhibit the cannibalistic behavior of late instars, allowing more larvae to survive and feed together in Bt ears [75,76,77]. If cannibalistic inhibition continues as Bt resistance increases, then a higher recruitment of *H. zea* adults could result from larvae surviving on a Bt plant compared to a non-Bt plant. This would have serious IRM implications for *H. zea* involving seed blends or structured refuges. However, we do not know how many larvae actually reach maturity in Cry1Ab ears, pupate, and successfully emerge as normal reproductive adults to contribute resistant alleles in the next generation. Nevertheless, given this worst-case scenario and an overall PFR of 0.99, the estimated RAF for Cry1Ab ranges somewhere between 0.495 (fully dominant resistance) and 0.995 (fully recessive resistance), assuming that *H. zea* resistance is based on a single autosomal locus. Simulation models indicate that the durability of the Cry1Ab toxin is basically lost or compromised when RAF exceeds 0.50 [58]. Given this high frequency of resistance alleles and the widespread decline in control efficacy against *H. zea*, most field corn hybrids expressing only Cry1Ab (events Bt11 and MON810) have been phased out of commercial use and replaced by pyramided Bt hybrids expressing multiple toxins. However, one remaining concern is that the cross resistance of Cry1Ab with other Cry toxins [18,47,78] may continue to reduce the durability of the pyramided hybrids.

Pyramided Bt corn expressing Cry1A.105/Cry2Ab2 toxins (MON 89034) was registered for use in 2010 [79] and initially provided the effective control of *H. zea* [80,81]. Since then, the PFRs of Cry1A.105/Cry2Ab2 in *H. zea* populations have steadily increased, averaging 0.19 during 2010–2013 and 0.41 during 2014–2016, according to the relative densities of surviving larvae in paired plots of Obsession I and Obsession II sweet corn in Maryland [29]. Starting in 2017, an expanded network of sentinel trials revealed further increased resistance, with PFRs averaging 0.67 in 2017, 0.93 in 2018, and 0.70 in 2019 [34]. Other laboratory and field studies conducted during this same time documented significant levels of resistance to Cry1A.105 and Cry2Ab2 in *H. zea* populations collected across the southeastern states [31,32,33,50,82,83]. In this study, levels of *H. zea* infestations and kernel damage were statistically lower in Cry1A.105/Cry2Ab2 ears, relative to levels in Cry1Ab ears, indicating that phenotypic resistance in *H. zea* has not yet reached the same level as resistance to Cry1Ab. However, our results show a consistent increase in kernel injury and the proportion of older instars surviving in Bt ears over the three years, indicating that *H. zea* continues to become less susceptible to the dual Cry toxins. In contrast with previous monitoring results [16,34], *H. zea* populations have evolved higher levels of phenotypic resistance to Cry1A.105/Cry2Ab2, as evident by PFR estimates averaging 0.88 in 2020, 0.93 in 2021, and 0.82 in 2022. The overall PFR was slightly higher at southern trial locations (mean = 0.90) but not statistically different from northern trials (mean = 0.87). Moreover, 32% of the trials since 2020 reported higher *H. zea* larval densities in Cry1A.105/Cry2Ab2 ears compared to non-Bt ears. These findings concur with recent studies reporting high resistance ratios and the increased field failure of the Cry1A.105 and Cry2Ab2 toxins in controlling *H. zea* infestations in Bt corn and Bt cotton [28,49,75,83,84,85]. As previously mentioned, it was not possible to estimate the RAF for each toxin, according to Venette et al. [60]. Studies using F2 screening methods have reported high levels of resistance alleles to individual Cry toxins in *H. zea* populations from southeastern states. Yu et al. [36] showed RAFs averaging 0.41 for Cry1A.105 and 0.33 for Cry2Ab2, while Santiago-Gonzalez et al. [53] estimated RAF to be 0.722 (95%CL: 0.688–0.764) for Cry1Ac and 0.217 (95%CL: 0.179–0.261) for Cry2Ab2. Altogether, given these studies, the high PFRs at most trial locations, and the reduced control efficacy in the field, the evidence clearly documents widespread *H. zea* resistance to Cry1A.105 and Cry2Ab2.

Currently, Vip3Aa pyramided with Cry toxins in Bt corn and Bt cotton provides high control efficacy against *H. zea*, and there is no evidence of practical resistance [39,49,50,51]. Studies during 2013–2016 in MD and WI found virtually no *H. zea* survival or damage in Vip3Aa sweet corn [29,48]. Additionally, industry registrants found no larval survival at high concentrations of Vip3Aa in diet bioassays on the 110 *H. zea* populations tested [58]. However, sentinel sweet corn monitoring during 2017–2019 reported some cases of larval survival in Vip3Aa ears with the expansion of trials to more southern locations [34]. During this time, 0.72% of the 9369 Vip3Aa ears sampled had minor tip damage associated with a small but noticeable increase in the number of surviving larvae. Similarly, our extended sentinel monitoring during 2020–2022 found mostly minor tip damage on 156 ears (0.77%) of the 20,312 ears sampled, of which 25 damaged ears were infested with live *H. zea* larvae. However, one notable difference from previous monitoring was that about one-half of the larvae found alive in Vip3Aa ears were late instars. Trials reporting most of the ear damage and older larvae in Vip3Aa ears were southern locations (TX, LA, MS, and AL). In particular, two trials in TX, two trials in AL, and one in MS recorded kernel consumption in 10% to 22% of the sampled Vip3Aa ears. However, not all of these damaged ears were tested for the presence of Vip3Aa, so there is the possibility that some ears resulted from contaminated non-Bt or Cry-expressing seed. At the same time, evidence of early stages of Vip3Aa resistance in *H. zea* populations in the southern states was reported from bioassay testing during 2016–2020 and from field reports of unexpected injury levels [49,50,51,52].

In this study, the overall worst-case estimate of PFR during 2020–2022 was 0.0042, assuming that all ears with live larvae produced Cry1Ab and Vip3Aa. This estimate is based on a total of 82 larvae found in 20,163 Vip3Aa ears compared to 10,682 larvae found in 11,622 non-Bt ears sampled. Further analyses by geographical region showed a higher PFR of 0.0133 averaged over 29 trials in the Gulf states, where the majority of live *H. zea* were found in Vip3Aa ears, in contrast to a PFR of 0.0010 averaged over the mid-Atlantic and North Central states, where 68 trials reported no *H. zea* survival in Vip3a ears. Assuming that resistance to Cry1Ab was nearly fixed in these trial locations and resistance to Vip3Aa was recessive [63], the overall estimated RAF in *H. zea* populations expressing resistance to Vip3Aa could range as high as 0.115 (95%CL: 0.107–0.124) in the Gulf states compared to an RAF of 0.0317 (95%CL: 0.0312–0.0323) at more northern trial locations. It is unlikely that these RAF values are underestimated because the resistance to Cry1Ab had no or little effect on variation in the PFR for Cry1Ab/Vip3Aa corn. On the other hand, as previously mentioned, the RAF estimates could be too high because we assumed that all surviving larvae found in Cry1Ab/Vip3Aa ears were resistant to Vip3Aa. Nevertheless, these RAFs range noticeably higher than the worst-case value of 0.02 used in simulation models by industry registrants to estimate the durability of the Vip3Aa toxin [85]. Other studies using F2 screen methods have reported lower estimates of RAF conferring Vip3Aa resistance, ranging from 0.0065 (95%CL: 0.0014–0.0157) in Texas populations [51] to 0.0155 (95%CL: 0.0057–0.0297) in populations from four southern states [53]. Using a group-mating approach, Lin et al. [37] also reported frequencies of major Vip3Aa resistance alleles of 0.028 and ranging from 0 to 0.0073 for minor resistance alleles. Taken together, these recent studies and our sentinel monitoring results show convincing evidence that the RAF for Vip3Aa in *H. zea* populations has been increasing since 2017, mainly in the southern states.

## 4. Conclusions

Our in-field monitoring network of sentinel trials provided information on the major lepidopteran pests in Bt and non-Bt corn over a large geographical area. Most importantly, the very low levels, or absence, of *O. nubilalis* in all trials further document the high-dose control efficacy of the Cry toxins and the areawide suppression of this pest by Bt corn. *Helicoverpa zea* infestations were very high in both non-Bt and Cry-expressing ears, particularly in southeastern and mid-Atlantic locations where overwintering occurs. Most disconcerting is that *H. zea* infestation levels in Cry1Ab and Cry1A.105/Cry2Ab2 ears were, respectively, similar and only slightly lower relative to their non-Bt isohybrids, with a third to one-half of the trials reporting higher larval densities in Cry-expressing ears compared to non-Bt ears. Our findings concur with many published studies that demonstrate the widespread field-evolved resistance to Cry toxins in *H. zea* populations. Unfortunately, the high resistance to Cry toxins might make it difficult for any regulatory mitigation action by the EPA or industry registrants to reduce or prevent further *H. zea* resistance to these toxins.

Vip3Aa pyramided with Cry toxins in Bt sweet corn, Bt field corn, and Bt cotton still provides excellent overall protection against *H. zea*. However, given the high levels of *H. zea* resistance to Cry toxins and the latter’s ineffectiveness against this pest, the redundancy control advantage of the pyramided Bt crops is compromised [17,18,19,85], which will likely lead to the faster evolution of resistance, especially when considering multiple generations of selection per season and the increased use of Vip3Aa field corn and cotton to improve the control of *H. zea* in the South. In sum, the time for proactive IRM measures for the Vip3Aa toxin is passing quickly, so we urgently need to implement best management practices to delay further Vip3Aa resistance, as outlined by Reisig et al. [46] and Gassmann and Reisig [67].

## Figures and Tables

**Figure 1 insects-14-00577-f001:**
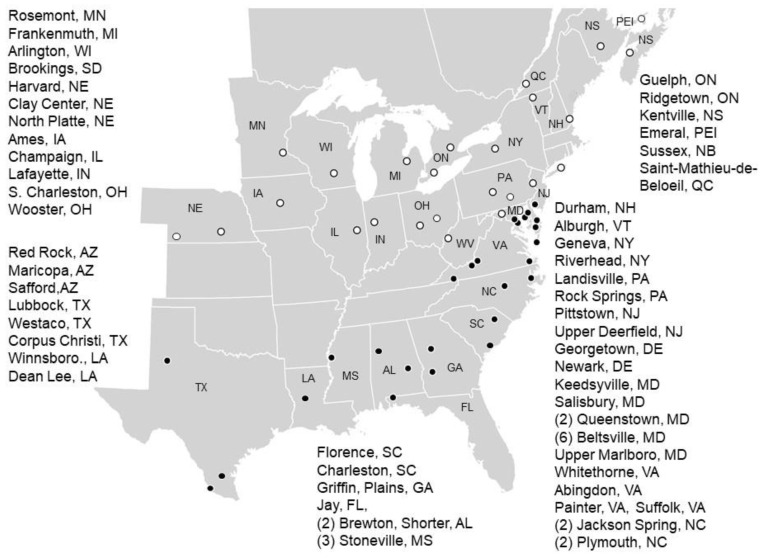
Trial locations of the sweet corn sentinel monitoring network during 2020–2022. Solid circles indicate locations where *Helicoverpa zea* overwinters, while open circles indicate where populations are mainly sourced by migrant moths from the south. Numbers in parentheses indicate multiple plantings of trials over the growing season. Not plotted are trials conducted in AZ and SD.

**Table 1 insects-14-00577-t001:** Mean (±SD) percentage of ears damaged by *Helicoverpa zea*, density of larvae per ear, kernel area consumed, and proportion of late larvae (4th, 5th, and 6th instars) in Bt sweet corn hybrids paired side-by-side to their non-Bt isolines. Means are listed by hybrid for each year ^a^.

Hybrid (Bt Trait)	Year [No. of Trials]	% Ears Damaged	Number of Larvae per Ear	Kernel Consumption per Damaged Ear (cm^2^)	Proportion of Late Instars
Obsession I (Non-Bt isoline to Obsession II)	2020 (41)	78.9 ± 30.3	1.13 ± 0.68	6.5 ± 3.8	74.0 ± 24.2
	2021 (42)	70.2 ± 31.9	0.88 ± 0.63	6.3 ± 3.5	78.0 ± 22.1
	2022 (44)	80.9 ± 30.5	1.20 ± 0.68	7.3 ± 4.0	79.1 ± 20.1
Obsession II (Cry1A.105 + Cry2Ab2)	2020 (41)	68.7 ± 29.6 ^#^	0.92 ± 0.68 ^#^	4.1 ± 2.8 ^#^	56.0 ± 28.1 ^#^
	2021 (46)	57.5 ± 36.8 ^#^	0.72 ± 0.61 ^#^	4.5 ± 3.3 ^#^	57.6 ± 32.1 ^#^
	2022 (45)	68.5 ± 33.0 ^#^	0.97 ± 0.72 ^#^	4.7 ± 3.2 ^#^	59.1 ± 32.5 ^#^
Providence (Non-Bt isoline to BC0805 and Remedy)	2020 (41)	80.0 ± 27.7	1.18 ± 0.71	7.0 ± 2.7	82.1 ± 17.5
	2021 (50)	70.6 ± 33.6	0.95 ± 0.68	6.2 ± 3.4	77.1 ± 22.7
	2022 (55)	80.1 ± 27.5	1.23 ± 0.70	7.3 ± 4.3	78.1 ± 18.3
BC0805 (Cry1Ab)	2020 (41)	77.1 ± 17.9	1.23 ± 1.02	5.4 ± 2.6 *	66.0 ± 27.7 *
	2021 (41)	66.7 ± 33.8	0.94 ± 0.70	5.6 ± 3.5	69.7 ± 24.9
	2022 (45)	74.9 ± 30.8	1.26 ± 1.01	6.0 ± 3.7	67.4 ± 23.3 *
Remedy (Cry1Ab + Vip3Aa)	2020 (41)	1.44 ± 4.09	0.004 ± 0.02	0.33 ± 0.88	0.0 ± 0.0
	2021 (52)	0.82 ± 2.18	0.006 ± 0.02	0.96 ± 2.05	10.1 ± 26.8
	2022 (53)	0.37 ± 0.99	0.004 ± 0.02	0.49 ± 1.27	6.9 ± 24.8

^a^ Data were averaged over all sentinel trials, except for trials with no *H. zea* infestation. ^#^ and * indicate significant differences between Obsession I and Obsession II hybrids and between Providence and BC0805 hybrids, respectively (paired *t*-test, *p* < 0.05). Remedy data were highly skewed and not statistically tested for differences from BC0805.

**Table 2 insects-14-00577-t002:** Mean phenotypic frequency of resistance (PFR) of *Helicoverpa zea* infestations and 95% confidence limits by sweet corn hybrid and monitoring year ^a^.

Hybrid (Bt Toxins Expressed)	Year	No. of Trials ^b^	Phenotypic Frequency of Resistance	95% Confidence Limits
BC0805 (Cry1Ab)	2020	31	0.97	0.85–1.09
	2021	28	0.98	0.84–1.07
	2022	33	1.02	0.95–1.09
Obsession II (Cry1A.105 + Cry2Ab2)	2020	23	0.88	0.77–0.98
	2021	23	0.93	0.78–1.08
	2022	31	0.82	0.75–0.90
Remedy (Cry1Ab + Vip3Aa)	2020	36	0.002	−0.0014–0.0048
	2021	37	0.009	0.0022–0.0153
	2022	43	0.002	0.0003–0.0048

^a^ PFR was computed for each trial as the ratio of the number of *H. zea* larvae surviving per Bt ear relative to larvae surviving per non-Bt isoline ear. ^b^ Means were averaged over sentinel trials each year that reported >50% of the Cry-expressing and non-Bt ears damaged and infested with >50% late *H. zea* instars.

## Data Availability

Data are available upon request.

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
