# Peer review of "Extended Sentinel Monitoring of *Helicoverpa zea* Resistance to Cry and Vip3Aa Toxins in Bt Sweet Corn: Assessing Changes in Phenotypic and Allele Frequencies of Resistance"

_insects, 2023, doi:10.3390/insects14070577_

Round 1
Reviewer 1 Report
This paper presents the results of sentinel monitoring of Helicoverpa zea resistance to Cry and Vip3Aa toxins in Bt sweet corn in 2020-2022 at sites in the USA and Canada by estimating changes in phenotypic and resistance allele frequencies. The work is a continuation of an earlier work by the same author. Still, this time more experimental sites were monitored, some of which contain multiple plantings, sampling was expanded to detect early resistance development to the Vip3Aa toxin, and ear sampling timing was improved to more accurately estimate phenotypic frequencies of resistance.
The topic is interesting and necessary as the development of Bt resistance in important pest species threatens Bt crop technology. The paper is impeccably written and well-structured. The conclusions are sound.
Minor comments:
Figure 1: It would be very descriptive if the authors could label each state (USA) / province (CA) with initials on the map, especially for non-U.S. / non-Canadian readers.
Line 253, P-6: Define GPD or GD
Author Response
Reply to Reviewer 1
As suggested, I added labels of the state and Canadian provinces that had trial locations to figure 1.
Regarding line 253, the GPD is the initials of the 1st author indicating the source of the unpublished data.
Reviewer 2 Report
Dively et al. present the field evidence of widespread resistance to Cry1Ab, Cry2Ab2, Cry1A.105 toxins in Helicoverpa zea. The study is impressive in its special extent and there are significant findings in the results, which I found highly suitable for the journal.
Still, there are a few things that might be considered by the authors to improve the readability and quality of the manuscript.
1. It confused me to understand the number of the states for multiple plantings at different times and/or locations (Lines 177-180).
2. The words “Highest infestations“ and “lowest infestations” are used to describe the occurrence and the pest population densities in different locations (Lines 207-310). It is understandable. However, the “ear infestations and larval densities” are used to describe the occurrence and the pest population densities in different Bt and non-Bt sweet corn hybrids (Lines 310-313). It confused me to think weather or not the pests have the behavior of host preference for egg laying between Bt and non-Bt sweet corn hybrids. As I know Bt corn control Lepidopteran insects by killing the larvae when they eat the Bt corn plant tissues. There would not be host preference by moths for egg laying between Bt and non-Bt corn hybrids. If the authors agree, then, the infestation levels of the insect pests should be the same or non-significantly different between Bt and non-Bt corn hybrids within a trial. In this case, I suggest the authors may use “damage” instead of “infestation”.
3. Both “isoline” and “isogenic hybrid” are used for presenting the similarity of Bt and non-Bt sweet corn hybrids used in the study. I suggest only use either isohybrid or isogenic hybrid, but not isoline, due to all sweet corn used in the study are hybrids, not line or inbred line.
4. Some text editing has been indicated in the manuscript. The numbers of references are disordered in the reference list. Check out for all species latin names, they should be italic.
Author Response
Reply to Reviewer 2
Reworded lines 177-182 to more clearly explain that most collaborators established one trial but there were other locations with multiple plantings and/or locations.
Regarding the use of the terms ‘highest and lowest infestations’, we agree that the corn earworm moths probably show no differential response to Bt or non-Bt expressing plants when ovipositing. However, it’s clear from the description of the methods and presentation of the results that we use the term infestation in specific reference to the percentage of ears infested with larvae and/or damage. So we see no reason to use the term damage in all situations, because ears can be infested with larvae but without damage.
Changed all mentions of isoline to isohybrid or isogenic hybrid.
All scientific names are given in italic.
References are in the proper order.